# Rifampicin tolerance and growth fitness among isoniazid-resistant clinical *Mycobacterium tuberculosis* isolates from a longitudinal study

Srinivasan Vijay[1,2,3,4†], Nguyen Le Hoai Bao[1†], Dao Nguyen Vinh[1], Le Thanh Hoang Nhat[1], Do Dang Anh Thu[1], Nguyen Le Quang[1], Le Pham Tien Trieu[1], Hoang Ngoc Nhung[1], Vu Thi Ngoc Ha[1], Phan Vuong Khac Thai[5], Dang Thi Minh Ha[5], Nguyen Huu Lan[5], Maxine Caws[6], Guy E Thwaites[1,2], Babak Javid[7], Nguyen Thuy Thuong[1,2*]

[1]Oxford University Clinical Research Unit, Ho Chi Minh, Viet Nam; [2]Centre for Tropical Medicine and Global Health, Nuffield Department of Medicine, University of Oxford, Oxford, United Kingdom; [3]Theoretical Microbial Ecology, Institute of Microbiology, Faculty of Biological Sciences, Friedrich Schiller University, Jena, Germany; [4]Cluster of Excellence Balance of the Microverse, Friedrich Schiller University Jena, Jena, Germany; [5]Pham Ngoc Thach Hospital, Ho Chi Minh, Viet Nam; [6]Department of Clinical Sciences, Liverpool School of Tropical Medicine, Liverpool, United Kingdom; [7]Division of Experimental Medicine, University of California, San Francisco, San Francisco, United States

*For correspondence:
thuongntt@oucru.org

†These authors contributed equally to this work

**Abstract** Antibiotic tolerance in *Mycobacterium tuberculosis* reduces bacterial killing, worsens treatment outcomes, and contributes to resistance. We studied rifampicin tolerance in isolates with or without isoniazid resistance (IR). Using a minimum duration of killing assay, we measured rifampicin survival in isoniazid-susceptible (IS, n=119) and resistant (IR, n=84) isolates, correlating tolerance with bacterial growth, rifampicin minimum inhibitory concentrations (MICs), and isoniazid-resistant mutations. Longitudinal IR isolates were analyzed for changes in rifampicin tolerance and genetic variant emergence. The median time for rifampicin to reduce the bacterial population by 90% (MDK90) increased from 1.23 days (IS) and 1.31 days (IR) to 2.55 days (IS) and 1.98 days (IR) over 15–60 days of incubation, indicating fast and slow-growing tolerant sub-populations. A 6 log10-fold survival fraction classified tolerance as low, medium, or high, showing that IR is linked to increased tolerance and faster growth (OR = 2.68 for low vs. medium, OR = 4.42 for low vs. high, p-trend = 0.0003). High tolerance in IR isolates was associated with rifampicin treatment in patients and genetic microvariants. These findings suggest that IR tuberculosis should be assessed for high rifampicin tolerance to optimize treatment and prevent the development of multi-drug-resistant tuberculosis.

## eLife assessment

This **valuable** study demonstrates that there is significant variation in the susceptibility of isoniazid-resistant Mycobacterium tuberculosis clinical isolates to killing by rifampicin, in some cases at the same tolerance levels as bona fide resistant strains. The evidence provided is **solid**, with no clear genetic marker for increased tolerance, suggesting that there may be multiple routes to achieving this phenotype. The work will be of interest to infectious disease researchers.

## Introduction

*Mycobacterium tuberculosis* causes around 10 million cases of tuberculosis (TB) each year and 1.5 million deaths (*WHO, 2021*). Challenges to successful TB treatment include bacterial evolution and diversification under host stresses and antibiotics, leading to differential antibiotic susceptibility even among genetically susceptible *M. tuberculosis* isolates (*Colangeli et al., 2018*). Based on killing dynamics, the differential susceptibility can be classified into two phenomena, (1) antibiotic tolerance observed as a reduced rate of killing of the entire bacterial population (*Kwan et al., 2013*), and (2) antibiotic persistence observed as a reduced rate of killing of sub-populations compared to more susceptible bacteria (*Brauner et al., 2016*; *Ronneau et al., 2021*). Clinically susceptible isolates exposed to host stresses and antibiotic selection can exhibit increased antibiotic tolerance and persistence (*Liu et al., 2016*; *Mishra et al., 2021*; *Gordhan et al., 2021*), as seen by the emergence of mutations increasing tolerance or persistence among clinical *M. tuberculosis* isolates (*Su et al., 2016*; *Torrey et al., 2016*; *Hicks et al., 2018*; *Wang et al., 2020*). Recent studies have also implicated the antibiotic tolerance in clinical isolates as a risk factor for hard-to-treat infections and tolerance can also contribute to the emergence of resistance (*Lee et al., 2019*) and relapse (*Imperial et al., 2018*).

Emergence of rifampicin tolerance or persistence, a key drug in TB treatment is a major concern considering the emergence of multi-drug resistant (MDR, resistant to at least isoniazid and rifampicin) tuberculosis (*Grobbelaar et al., 2019*). Several mechanisms lead to rifampicin tolerance, heteroresistance, or persistence (*Adams et al., 2021*). These include efflux pump overexpression (*Adams et al., 2011*), mistranslation (*Javid et al., 2014*), overexpression of rifampicin target *rpoB* (*Zhu et al., 2018*), cell size heterogeneity, *Vijay et al., 2017*; *Aldridge et al., 2012*; *Rego et al., 2017* and the redox heterogeneity in bacteria (*Mishra et al., 2019*). Rifampicin treatment can also result in differentially detectable sub-populations of *M. tuberculosis*, which can grow only in liquid medium as compared to solid medium (*Saito et al., 2017*). Therefore, in determining the risk of post-treatment relapse, it is important to consider, alongside tolerance range, the degree of growth heterogeneity within tolerant subpopulations.

Apart from rifampicin susceptibility variation, another concern in standard TB treatment is the emergence of IR. There is globally around 10% prevalence of IR among clinical *M. tuberculosis* isolates (*Thai et al., 2018*). IR is difficult to rapidly diagnose during drug susceptibility testing, and is associated

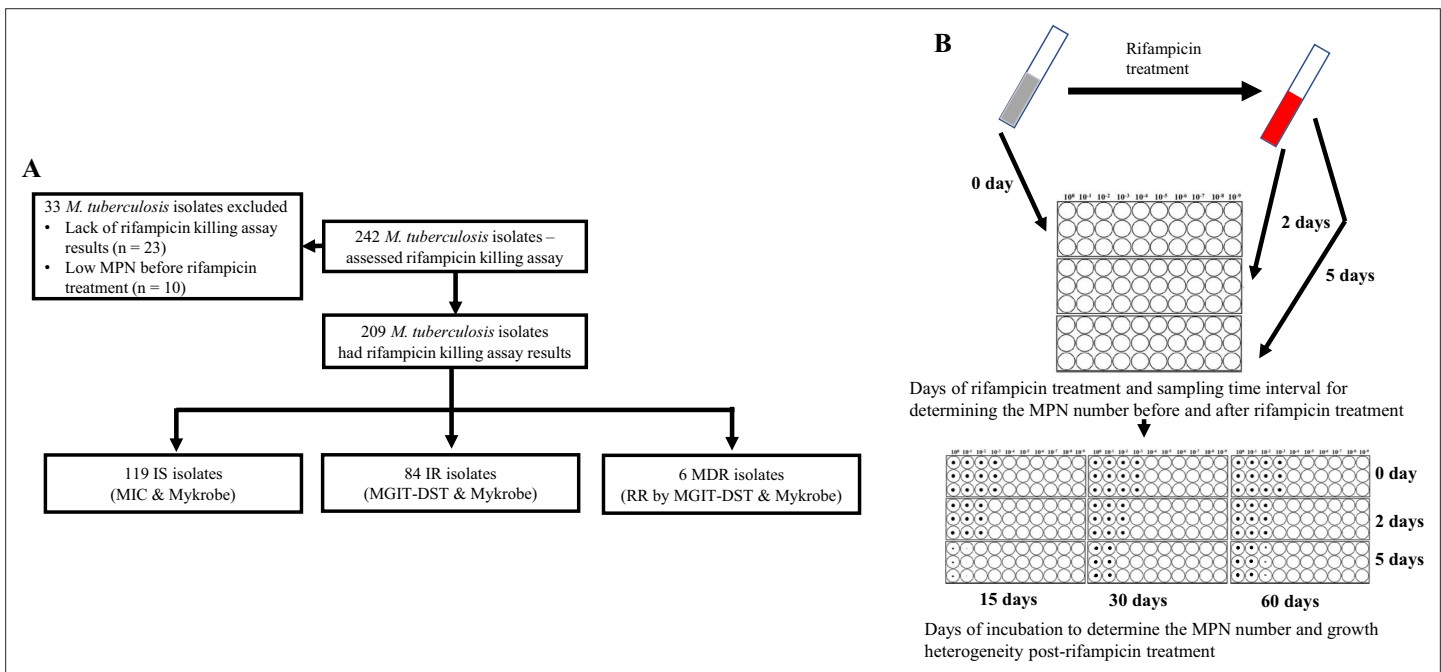

**Figure 1.** Study design. (**A**) Study design. IS – Isoniazid susceptible, IR – Isoniazid-resistant, RR – Rifampicin-resistant. (**B**) Most-probable number-based rifampicin killing assay and survival fraction determination.

with worse treatment outcomes compared to isoniazid-susceptible (IS) *M. tuberculosis* isolates (*Thai et al., 2018*). Importantly, IR is also associated with the subsequent emergence of rifampicin resistance leading to MDR TB (*Srinivasan et al., 2020*).

Despite its potential importance for the TB treatment, the distribution of rifampicin tolerance among clinical *M. tuberculosis* isolates is unknown, and routine clinical microbiology diagnosis does not include any assays for tolerance. The growth fitness of rifampicin tolerant subpopulations, and the association of pre-existing IR with rifampicin tolerance is completely unknown.

To address this knowledge gap, we developed a most-probable number (MPN) based minimum duration of killing (MDK) assay to determine the rifampicin tolerance among clinical *M. tuberculosis* isolates in a medium-throughput manner (*Vijay et al., 2021*). In the current study, we investigated the rifampicin tolerance in a large set of IS (n=119) and IR (n=84) clinical *M. tuberculosis* isolates and its correlation with bacterial growth rate, rifampicin MICs, IR mutations, and the rifampicin treatment selection in patients.

## Results

### Study design

We investigated rifampicin tolerance and its association with isoniazid susceptibility among 242 clinical *M. tuberculosis* isolates. We treated susceptible isolates with rifampicin (2 µg/mL), a concentration several times higher than their MICs (*Supplementary file 1*) and also close to the serum rifampicin concentration observed in a patient during oral dose (*Prakash et al., 2003*), and at 0, 2, and 5 days determined fractional survival following 15, 30, and 60 days of culture (*Figure 1A*). Higher survival fractions indicate higher rifampicin tolerance, and differences in survival fraction determined between 15 and 60 days of incubation indicated greater growth heterogeneity in rifampicin tolerant sub-populations (*Figure 1B*). 23 of the isolates grew poorly in the absence of antibiotics, and a further 10 had low initial MPN, making accurate determination of survival fractions difficult (*Figure 1A*), and these 33 isolates were removed from further analysis. Of the remaining 209 isolates, 119 IS, 84 IR, and 6 were resistant to both rifampicin and isoniazid, MDR. The MDR isolates were controls and comparators as isolates with a known high degree of rifampicin tolerance (*Vijay et al., 2021*).

### Distribution of Rifampicin tolerance in IS and IR isolates

We analyzed the rifampicin survival fraction and the kill curve for IS and IR *M. tuberculosis* isolates, at 0, 2, and 5 days of rifampicin treatment followed by 15 and 60 days of incubation (*Figure 2*). We did not further analyze 30 days incubation result, as it was similar to 60 days incubation (*Figure 2—figure supplement 1*). Following 5 days of rifampicin treatment, the average survival fraction reduced by 90–99% of the starting bacterial population (*Figure 2*). We calculated the time required for 90% survival fraction reduction ($MDK_{90}$) for each isolate by determining the different lengths of the X-axis (Days post rifampicin treatment) corresponding to a 90% decline in survival fraction in the Y-axis (*Figure 2*, *Figure 2—figure supplement 2*, and *Figure 2—figure supplement 3*). Of note, the average time required for 90% survival fraction reduction ($MDK_{90}$) was 1.23 (95%CI (Confidence interval) 1.11; 1.37) and 1.31 (95%CI 1.14; 1.48) days for IS and IR, respectively when survivors were incubated for 15 days, but rose to 2.55 (95%CI 2.04; 2.97) and 1.98 (95%CI 1.69; 2.56) days for 60 days for IS and IR isolates, respectively (*Figure 2*). This shift in the $MDK_{90}$ indicated the presence of growth heterogeneity within the tolerant subpopulation – with both fast and slow-growing bacteria within tolerant subpopulations. For most of the isolates, $MDK_{90}$ time could be calculated but other parameters of tolerance and persistence such as $MDK_{99}$ (at 15 day = 81% (170/209), 60 day = 41% (86/209)) and $MDK_{99.99}$ (at 15 day = 11% (22/209), 60 day = 8% (17/209)) could be calculated for only a fraction of 209 isolates and the rest were beyond the assay limits (*Figure 2—figure supplement 2*). Intriguingly, we observed a significant difference in rifampicin tolerance between IS and IR isolates at 5 days of treatment– but only in the 15 days post-recovery. The difference had disappeared by 60 days (*Figure 2*). Therefore, we decided to consider survival fractions with 15 and 60 days recovery for further analysis, the earliest and latest time points for determining the fast- and slowly-growing rifampicin-tolerant subpopulations.

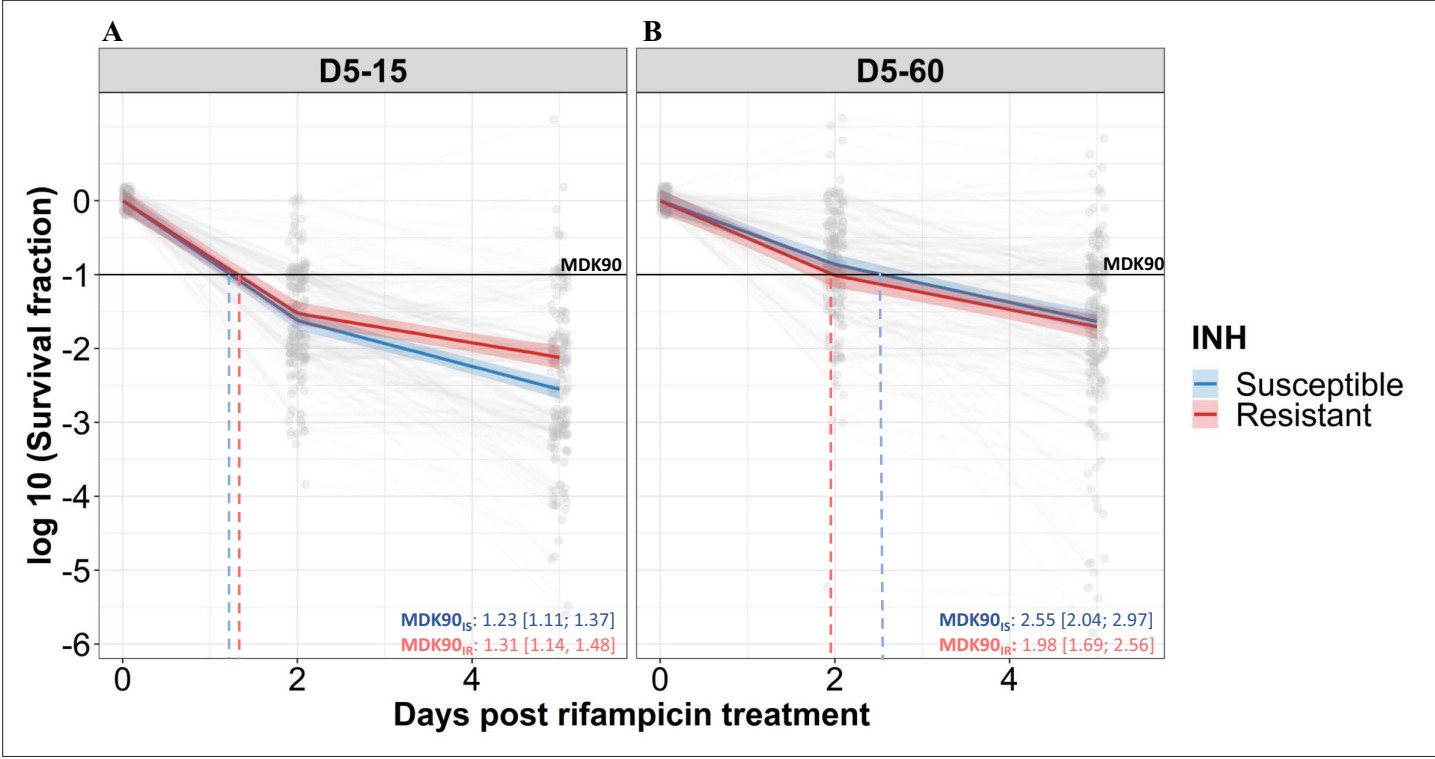

**Figure 2.** Rifampicin survival curve in isoniazid susceptible and resistant clinical *M. tuberculosis* isolates. (**A, B**) The bacterial kill curve as measured by log10 survival fraction from data collected at 0, 2, and 5 days of rifampicin treatment followed by incubation for 15 and 60 days, respectively. Data from individual isolates are shown as gray dots connected by lines. Estimated mean with 95% credible interval (bold coloed line and color shaded area, respectively) of isoniazid susceptible (IS, Isoniazid susceptible – blue, n=119, 117 for 15 and 60 days of incubation, respectively) and resistant (IR, Isoniazid-resistant – red, n=84, 80 for 15 and 60 days of incubation, respectively) clinical *M. tuberculosis* isolates based on linear mixed effect model implemented in a Bayesian framework. One log10 fold or 90% reduction in survival fraction is indicated (MDK90, black horizontal line) and the mean time duration required for 90% reduction in survival (MDK90, minimum duration of killing time) at 15 and 60 days of incubation is indicated by vertical dashed lines with respective colors for IS and IR isolates.

The online version of this article includes the following figure supplement(s) for figure 2:

**Figure supplement 1.** Rifampicin survival curve in isoniazid susceptible and resistant clinical *M. tuberculosis* isolates.

**Figure supplement 2.** Distribution of MDK$_{90, \, 99,}$ and $_{99.99}$ time (in days) for isoniazid susceptible (IS) and resistant (IR) isolates at 15 and 60 days incubation.

**Figure supplement 3.** Flowchart for calculating MDK 90, 99, and 99.99 time for clinical *M. tuberculosis* isolates.

## Isoniazid resistance is associated with fast-growing rifampicintolerant subpopulations

To further group rifampicin tolerance level, and correlate it with growth fitness and isoniazid susceptibility, we compared the distribution of survival fraction at 15 and 60 days recovery following 2 and 5 days of rifampicin treatment in IS (n=119) and IR (n=84) isolates (***Figure 3A***, ***Figure 3—figure supplement 1***). There was no significant difference in rifampicin tolerance between IS and IR isolates at 2 days of treatment (***Figure 3—figure supplement 1***). At 5 days of rifampicin treatment and both early (15 days) and late (60 days) recovery time points, IS and IR isolates showed a broad distribution of fractional survival–spanning 1 million times difference in rifampicin susceptibility (***Figure 3A***). At the 15 days recovery period, IR was significantly associated with higher survival to rifampicin treatment as compared to IS isolates (p=0.017, ***Figure 3A***), whereas at 60 days, fractional survival increased in both groups with no difference according to isoniazid susceptibility (***Figure 3A***). These results suggest that the difference between IS and IR rifampicin tolerant subpopulations is within their fast-growing tolerant bacilli only.

To further refine the distribution of rifampicin tolerance between isolates, first, we combined the rifampicin survival fraction distribution of both IS and IR isolates, then the fractional rifampicin survival

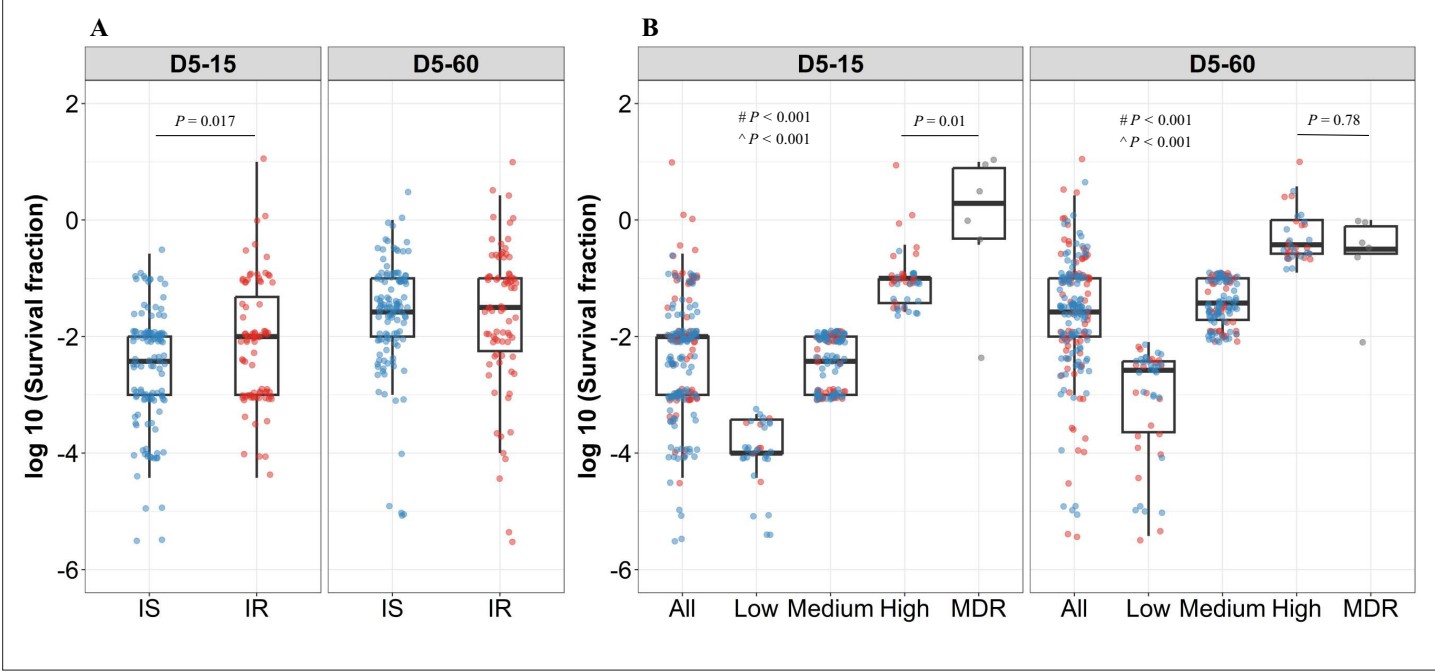

**Figure 3.** Rifampicin survival fraction distribution in isoniazid susceptible and resistant clinical *M. tuberculosis* isolates. (**A**) Log10 rifampicin survival fraction distribution, with median and IQR (interquartile range), of individual isoniazid susceptible (IS, blue dots, n=119, 117 for D5-15, and D5-60, respectively), and resistant (IR, red dots, n=84, 80 for D5-15, D5-60, respectively) isolates for 5 days of rifampicin treatment as determined at 15 and 60 days of incubation (D5-15, D5-60, respectively). (**B**) Rifampicin tolerance distribution in both IS (blue dots) and IR (red dots) isolates combined together (All) was used to group them as low (<25th percentile, n=33, 47 for D5-15, and D5-60, respectively), medium (from 25th to 75th percentile, n=124, 115 for D5-15, and D5-60, respectively) and high (above 75th percentile, n=46, 35 for D5-15, and D5-60, respectively) level of rifampicin tolerance and compare it with rifampicin tolerance of multi-drug resistant (MDR) clinical *M. tuberculosis* isolates (gray dots, n=6), after 5 days of rifampicin treatment and determined at 15 and 60 days of incubation (D5-15, D5-60, respectively). Statistical comparisons between Low, Medium, and High or MDR were made by using the Wilcoxon rank-sum test. # p-value for comparing the Low and High tolerance groups, ^ p-value for comparing the medium and High tolerance groups.

The online version of this article includes the following figure supplement(s) for figure 3:

**Figure supplement 1.** Log10 survival fraction distribution in isoniazid susceptible (IS) and resistant (IR) clinical *M. tuberculosis* isolates post 2 days of rifampicin treatment at 15 (D2-15) and 60 (D2-60) days of incubation.

was parsed as low, medium, or high as defined by falling within the 25th, 75th, and 100th percentiles of survival fractions following rifampicin treatment and either 15 or 60 days recovery (*Figure 3B*). As expected, there was substantially lower tolerance to rifampicin in low and medium groups compared with MDR isolates. Surprisingly, tolerance to rifampicin between non-rifampicin resistant 'high' tolerance strains and MDR strains was not significantly different (p=0.78, *Figure 3B*), and these high tolerant strains were characterized in both IS and IR isolates. This suggests that within the IR, high tolerant subgroup, antibiotic susceptibility (to both rifampicin and isoniazid) may be similar to *bona fide* MDR strains.

Analyzing rifampicin tolerance subgroups between IS and IR strains, at the early, 15- day recovery time-point, the majority (79%, 26/33) of 'low' rifampicin tolerant strains were isoniazid susceptible. By contrast, IR isolates were over-represented in the 'medium' and 'high' tolerant subgroups (OR of 2.7 and 4.4, respectively–*Table 1*). These associations disappeared with longer (60- day) recovery post-antibiotic treatment, confirming that IR isolates harbored fast-growing, high-level rifampicin-tolerant bacilli compared with IS isolates (*Table 1*).

**Table 1.** Association of rifampicin tolerance level with isoniazid susceptibility.

| Incubation time | Rifampicin tolerance level | Isoniazid susceptible (n=119) | Isoniazid resistant (n=84) | p | OR (95% CI) | p-trend |
|---|---|---|---|---|---|---|
| D5-15 | Low tolerance (n, %) | 26 (79, 26/33) | 7 (21, 7/33) | | | 0·0038 |
| | Medium tolerance (n, %) | 72 (58, 72/124) | 52 (42, 52/124) | 0·029 | 2·68 (1·08–6·65) | |
| | High tolerance (n, %) | 21 (46, 21/46) | 25 (54, 25/46) | 0·003 | 4·42 (1·60–12·22) | |
| D5-60 | Low tolerance (n, %) | 26 (55, 26/47) | 21 (45, 21/47) | | | 0·67 |
| | Medium tolerance (n, %) | 74 (64, 74/115) | 41 (36, 41/115) | 0·28 | 0·69 (0·34–1·37) | |
| | High tolerance (n, %) | 17 (49, 17/35) | 18 (51, 18/35) | 0·55 | 1·31 (0·55–3·15) | |

n = number of isolates. (% as percentage), N/total number (IS + IR). p = p-value determined using Chi-square test. p trend = p-value determined using Cochran-Armitage test. p trend = p-value determined using the Cochran-Armitage test. OR = odds ratio. 95%CI = 95% confidence interval.

## Association between rifampicin tolerance and relative growth in the absence of antibiotics, rifampicin MICs, isoniazid-resistant mutations of *M. tuberculosis* isolates

Clinical isolates of *M. tuberculosis* exhibit a large degree of lag time and growth heterogeneity (*von Groll et al., 2010*), as well as differences in rifampicin MICs or isoniazid-resistant mutations. Prior studies showed that slow growth rate and non-replicating persistence were correlated (*Pontes and Groisman, 2019*), therefore, we wished to investigate the association between growth rates in the absence of antibiotic treatment, rifampicin MIC distribution, isoniazid-resistant mutations, and rifampicin tolerance distribution in *M. tuberculosis* isolates.

For correlating relative growth in the absence of antibiotics, we removed 19 outliers which deviated from normal distribution (*Figure 4—figure supplement 1* with 19 outliers), Intriguingly, slower growth before rifampicin treatment did not have a significant the correlation with higher growth fitness in rifampicin survival fraction at 15 days incubation in IS isolates (*Figure 4A* regression coefficient –0.21, 95% CI [–0.44, 0.007], p=0.058). By contrast, the correlation of slower growth with lower growth fitness in the long recovery period was observed in both IS and IR isolates (*Figure 4B*, regression coefficient for IS = 0.38 [0.15, 0.61], p=0.0014, and IR = 0.38 [0.12, 0.64], p=0.0041). Comparing IS and IR isolates, IR isolates had slower growth in the absence of antibiotics (*Figure 4C*, p<0.0001). Thus, slow growth before rifampicin treatment correlates with reduced growth fitness in certain rifampicin tolerant populations at 60 days incubation.

In case of IS isolates, higher rifampicin MICs correlated with lower rifampicin tolerance at long recovery period, 15 (-0.24 [–0.50, 0.022], p=0.073) and 60 days incubation (–0.31 [-0.53,–0.083], p=0.007, *Figure 4—figure supplement 2*), whereas IR isolates did not show such a negative correlation of rifampicin tolerance with rifampicin MICs (0.14 [-0.14, 0.41], p=0.33 and 0.21 [-0.057, 0.48], p=0.12, *Figure 4—figure supplement 2*). This latter observation might be due to the increased growth fitness of IR rifampicin tolerant populations. In addition, there was no significant difference in rifampicin MICs distribution between IS and IR isolates (*Figure 4—figure supplement 3*).

We next investigated the association between isoniazid-resistant mutations in *M. tuberculosis* isolates and rifampicin tolerance distribution. These isolates had three different isoniazid-resistant mutations, *katG*_S315X (n=71), *inhA*_I21T (n=2), and *fabG1*_C-15X (n=6), and data not available for five isolates (*Figure 4—figure supplement 4*). Due to a low number of isolates with inhA and fabG1 mutations, it was not possible to identify the difference in rifampicin tolerance distribution between the isolates with different isoniazid-resistant mutations. Nevertheless, we observed wide distribution of rifampicin tolerance in isoniazid-resistant isolates with katG_S315X mutation itself (*Figure 4—figure*

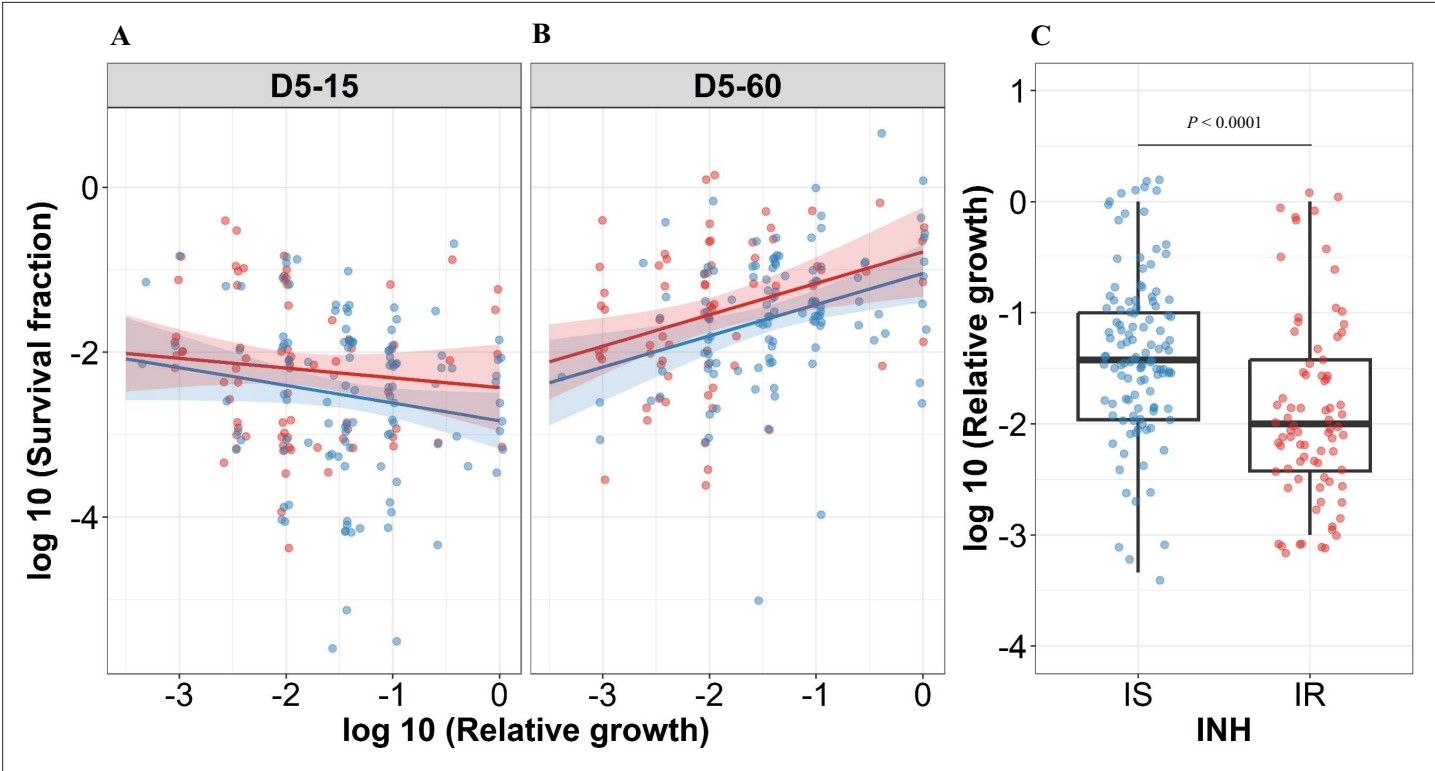

**Figure 4.** Correlating rifampicin survival fraction with before treatment relative growth of clinical *M. tuberculosis* isolates. Log10 survival fraction at 5 days of rifampicin treatment as determined at 15 days (**A**) and 60 days of incubation (**B**), for isoniazid susceptible (IS, blue dots) and resistant (IR, red dots) isolates, respectively, correlated with the log10 relative growth determined before rifampicin treatment for clinical *M. tuberculosis* isolates. Coefficients of linear regression for (**A**) IS = −0.21 [-0.44, 0.007], p=0.058; IR = −0.12 [-0.38, 0.14], p=0.37, and (**B**) IS = 0.38 [0.15, 0.61], p=0.0014; IR = 0.38 [0.12, 0.64], p=0.0041. (**C**) Log10 distribution of relative growth with median and interquartile range (IQR) for IS and IR clinical *M. tuberculosis* isolates before rifampicin treatment. Statistical comparisons between IS and IR were made by using the Wilcoxon rank-sum test.

The online version of this article includes the following figure supplement(s) for figure 4:

**Figure supplement 1.** Correlating rifampicin survival fraction with before treatment relative growth of clinical *M. tuberculosis* isolates with outliers included.

**Figure supplement 2.** Correlating rifampicin survival fraction with rifampicin minimum inhibitory concentration (MIC) of clinical *M. tuberculosis* isolates.

**Figure supplement 3.** Rifampicin minimum inhibitory concentration (MIC) distribution between Isoniazid susceptible (IS) (n=119) and Isoniazid-resistant (IR) (n=67) clinical *M. tuberculosis* isolates.

**Figure supplement 4.** Rifampicin tolerance distribution grouped based on isoniazid-resistant mutations (katG_S315X, inhA_I21T, and fabG1_C-15X) in *M. tuberculosis* isolates.

supplement 4), indicating the role of other genetic or epigenetic determinants influencing rifampicin tolerance.

## Higher rifampicin tolerance and growth fitness is associated with IR isolates from the intensive phase of treatment with rifampicin

The IS isolates were collected only at baseline before treatment, whereas the IR isolates in our study were collected longitudinally from patients at different stages of treatment. Both patients with IS and IR isolates received the standard 8 months treatment regimen according to the Vietnamese National TB Program during the study period (*Thai et al., 2018*), this included an initial two months with four antibiotics (streptomycin or ethambutol, with rifampicin, isoniazid, and pyrazinamide) followed by 6 months with isoniazid and ethambutocl (*Thai et al., 2018*). The antibiotic treatment may select different *M. tuberculosis* genetic micro variants in the patients and lead to differences in rifampicin tolerance between longitudinal isolates. Therefore, we analyzed the rifampicin tolerance distribution in the IR isolates in three sub-groups, before treatment (IR-BL), initial two months of intensive phase of treatment with rifampicin in the regimen (IR-IP), continuous phase, and relapse lacking rifampicin and

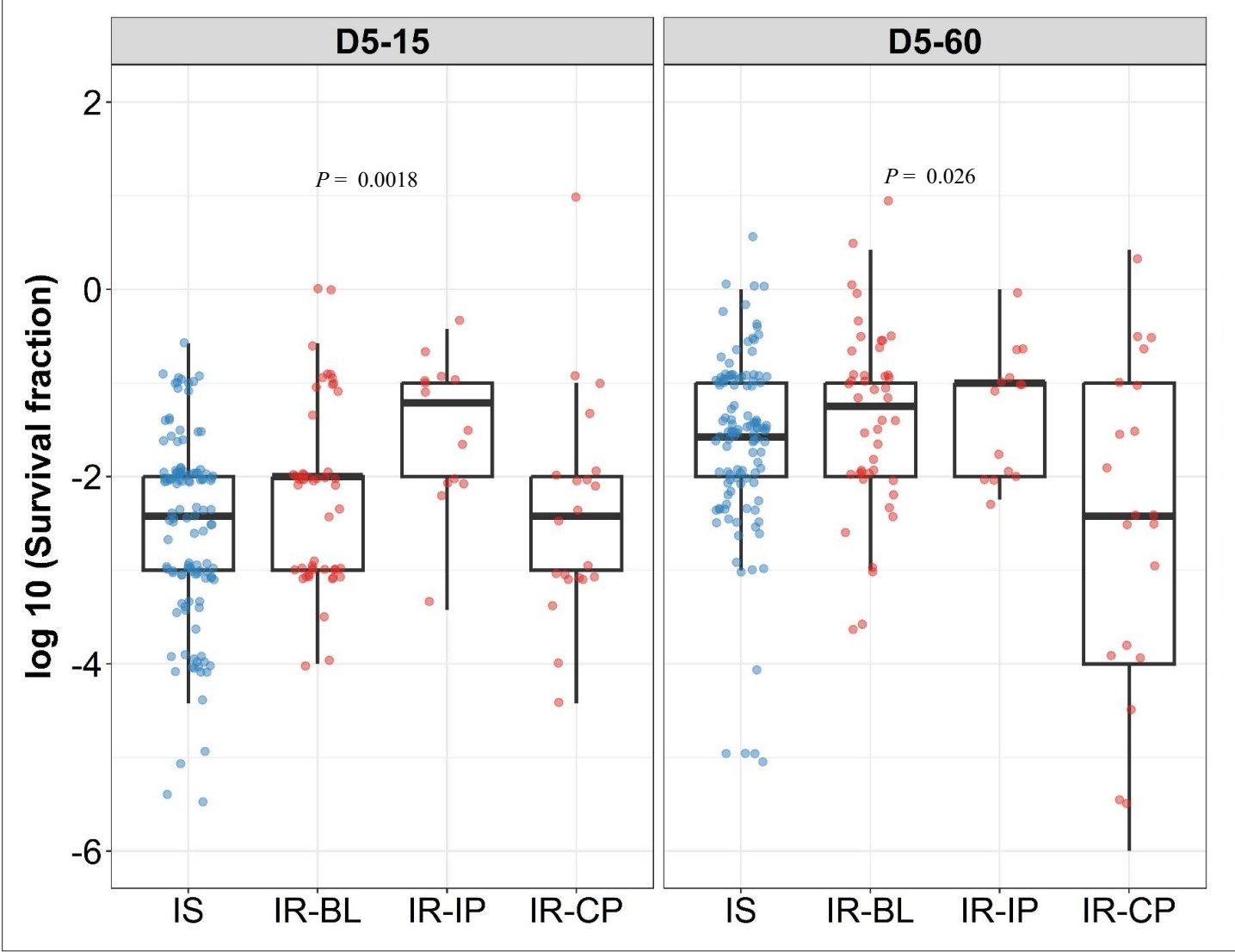

**Figure 5.** Rifampicin survival fraction distribution in isoniazid susceptible and longitudinal isoniazid-resistant clinical *M. tuberculosis* isolates. Log10 rifampicin survival fraction distribution, with median and IQR (interquartile range), of individual isoniazid susceptible (Isoniazid susceptible, IS, blue dots, n=119, 117 for D5-15, and D5-60, respectively), and longitudinal isoniazid-resistant (Isoniazid-resistant, IR, red dots, n=84, 80 for D5-15, D5-60, respectively) isolates for 5 days of rifampicin treatment as determined at 15 and 60 days of incubation (D5-15, D5-60, respectively) grouped based on collection time as baseline (IR-BL, n=49), intensive phase (IR-IP, n=14), and continuous phase and relapse (IR-CP, n=21). Statistical comparisons between groups were made by using Krusal-Walis test.

any other antibiotics treatment selection, respectively (IR-CP) (*Figure 5*). This grouping the reflects TB-treatment regimen in Vietnam during the study period with rifampicin only in the initial two months of treatment (*Thai et al., 2018*).

Interestingly, we observed significantly higher rifampicin tolerance and growth fitness in IR-IP group p=0.0018, *Figure 5* as compared to IS, IR-BL, and IR-CP groups during 15 days of recovery, indicating rifampicin treatment itself as a possible mechanism leading to the selection of *M. tuberculosis* tolerant microvariants in patients (*Zhu et al., 2018*).

To verify this finding, we grouped individual patients (n=18) based on changes in rifampicin tolerance between their initial and subsequent IR isolates collected before treatment (0 months), during treatment (1–8 months), and post-treatment (12–24 months) (*Figure 6*). We observed three kinds of changes in rifampicin tolerance between the isolates collected from the same patient, (1) decrease (one or more subsequent isolates with lower rifampicin tolerance as compared to the initial isolate), (2) unchanged (initial and subsequent isolates with similar level of rifampicin tolerance) and (3) Increase

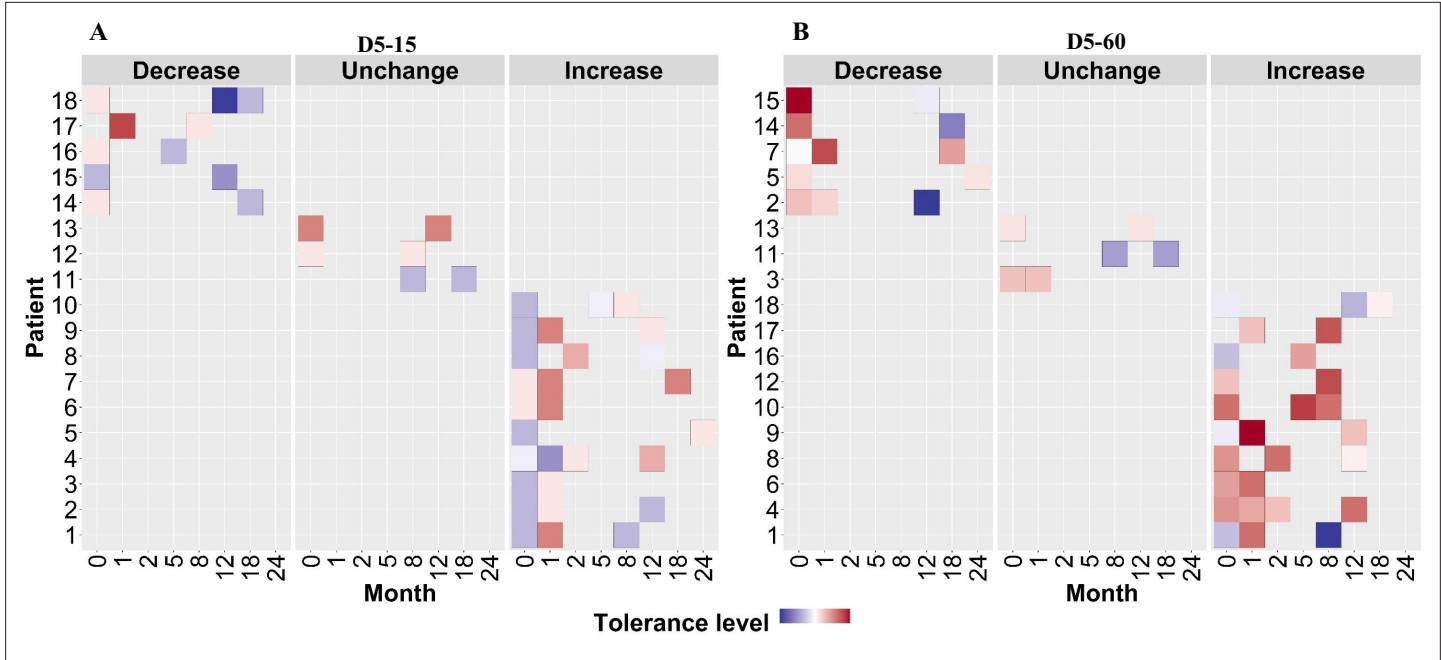

**Figure 6.** Rifampicin tolerance of longitudinal isoniazid-resistant clinical *M. tuberculosis* isolates from individual patients. (**A, B**) Rifampicin tolerance heat map after 5 days of rifampicin treatment as determined at 15 and 60 days of incubation (D5-15, D5-60, respectively), of longitudinal isoniazid-resistant clinical *M. tuberculosis* isolates collected from individual patients during different months of treatment and follow-up. Longitudinal isoniazid-resistant clinical *M. tuberculosis* isolates from individual patients are grouped based on changes in rifampicin tolerance compared between initial and subsequent months of collection as decrease, un change, and increase. Months (0–24) represent the different months the isolates were collected from patients during 8 months treatment and 24 months of follow-up.

(one or more subsequent isolates with higher rifampicin tolerance as compared to the initial isolate) for 5 days or rifampicin treatment and 15 and 60 days recovery time (*Figure 6*) and analyzed the difference in non-synonymous SNPs between the isolates from the same patients associated with differences in rifampicin tolerance (*Figure 7*, *Supplementary file 1b*). The SNPs difference between the longitudinally collected *M. tuberculosis* isolates from the same patient were 0–3 except in one case (SNPs = 11), indicating de-novo emergence or selection of genetic microvariants within the patient (*Supplementary file 1*). Next, we analyzed the non-synonymous SNPs associated with the changes in rifampicin tolerance both at 15 and 60- days incubation. This included both genetic variants emerging as more than 90% of WGS reads and less than 90% threshold used as a cut-off for calling SNPs. Several genes Rv0792c, Rv1266c, Rv1696, Rv1758, Rv1997, Rv2043c, Rv2329c, Rv2394, Rv2398c, Rv2400c, Rv2488c, Rv2545, Rv2689c, Rv3138, Rv3680, and Rv3758c previously reported to be associated with persistence, tolerance and survival within host had non-synonymous SNPs associated with changes in rifampicin tolerance (*Figure 7*, *Supplementary file 1c* with references). This indicates mutations in multiple genes might affect rifampicin tolerance and growth fitness, since there was no one gene or genetic variant in *M. tuberculosis* in multiple patients consistently associated with increased or decreased rifampicin tolerance, or that mutations may be epistatic with the genetic background of the strain.

## Discussion

We investigated rifampicin tolerance in a large number of clinical isolates of *M. tuberculosis*. Overall clinical *M. tuberculosis* isolates showed higher levels of rifampicin tolerance than lab isolates as the average survival fraction post-rifampicin treatment decreased only by 90–99% over 5 days. We found that levels of rifampicin tolerance are widely distributed among isolates, with some genetically susceptible strains having similar susceptibility to rifampicin-mediated killing as *bona fide* rifampicin-resistant

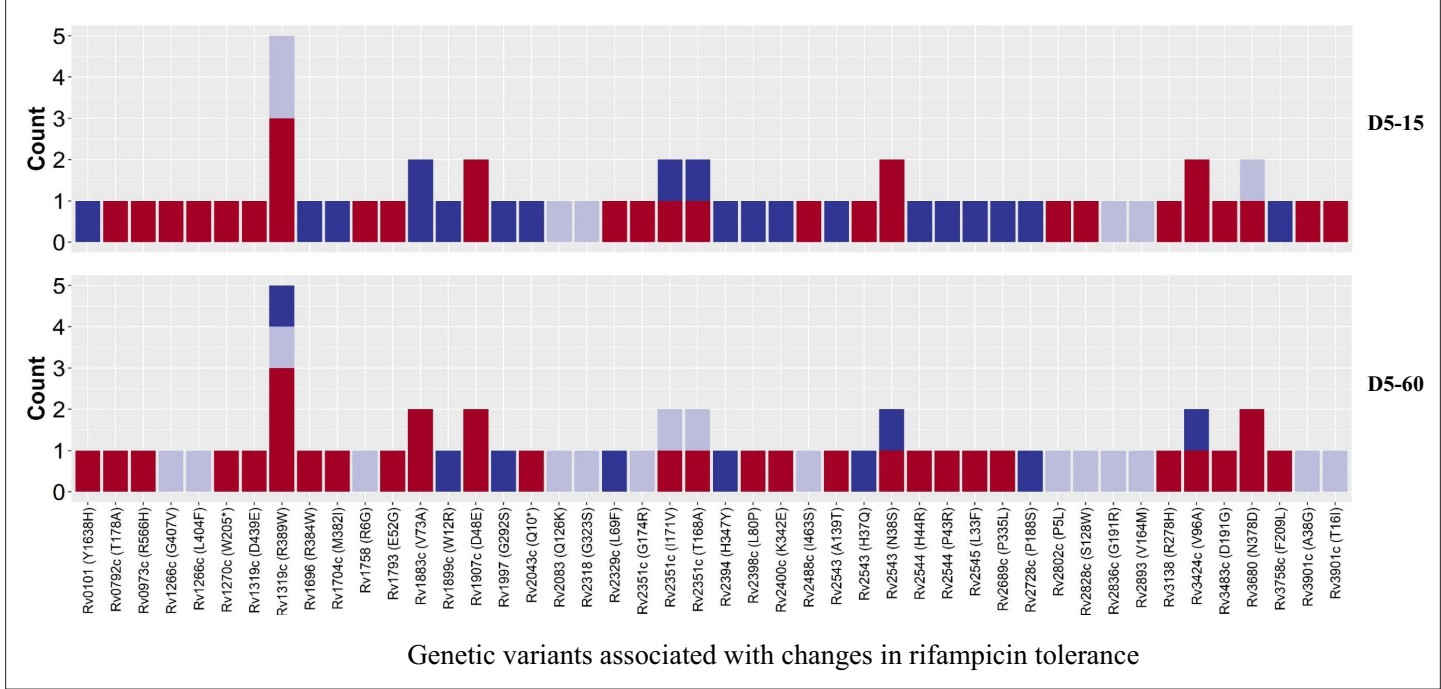

**Figure 7.** Genetic variants associated with changes in rifampicin tolerance. Non-synonymous single nucleotide polymorphism emerging in pair-wise comparison of longitudinally collected isoniazid-resistant *M. tuberculosis* isolates from same patient associated with increase (red), decrease (dark blue), and no change (light violet) in rifampicin tolerance phenotype at 15 and 60 days of incubation (D5-15 and D5-60, respectively). Each count represents a single independent single nucleotide polymorphism (SNP) emergence event.

isolates, at least during the 5 days of rifampicin exposure of our assay condition. Furthermore, IR isolates were more likely to harbor fast-growing subpopulations with high levels of rifampicin tolerance.

Heterogeneity in regrowth following stress has been linked to a tradeoff between growth fitness and survival (*Moreno-Gámez et al., 2020*), and it is likely that in *M. tuberculosis* such diversification in growth rates among rifampicin-tolerant subpopulations represents such a balance between growth and persistence under antibiotic stress.

We also observed a variation in growth rate in the absence of antibiotic therapy. On average, IR isolates were slower growing than IS isolates, which likely represents a fitness cost due to isoniazid- resistance-causing mutations and strain genetic background (*Gagneux, 2009*). As expected, IS isolates, with slower growth in the absence of a drug had a weak association with high levels of rifampicin tolerance at the 15- day time point (*Pontes and Groisman, 2019*) (representing rapidly growing recovered cells), whereas both IS and IR isolates with slower growth in the absence of drug were significantly associated with lesser rifampicin survival fraction levels at 60 days– representing slow growing rifampicin tolerant bacilli. These data suggest that slower growth (in absence of a drug) in both isoniazid susceptible and resistant isolates, perhaps due to the fitness cost of mutations (*Gagneux, 2009*), may be associated with more persister-like tolerant subpopulations.

By contrast, the association between rifampicin MIC and rifampicin tolerance showed a contrasting trend with isoniazid susceptibility. IS isolates showed decreased tolerance with the increase in rifampicin MIC, but IR isolates did not show this association. This may indicate higher growth fitness of IR with rifampicin tolerance. Another important finding from our study is the emergence of higher rifampicin tolerance and growth fitness in longitudinal IR isolates under rifampicin treatment selection. This further supports the findings that multiple genetic microvariants may co-exist in patients and rapidly change their proportion under selection from host stresses and antibiotic treatment (*Trauner et al., 2017*). We also observed non-synonymous mutations in multiple genes, associated with persistence and host survival enriched with changes in rifampicin tolerance between the longitudinal isolates (*Supplementary file 1c* with references). However, the lack of convergent SNPs in the samples may be due to the relatively small sample size, interaction between SNPs, and strain background, or indication

of a larger set of tolerance-related genes that independently affect bacterial growth and antibiotic tolerance (*Brauner et al., 2016*).

Our study also reveals novel aspects of rifampicin tolerance associated with isoniazid susceptibility. Rifampicin treatment itself led to the selection of IR *M. tuberculosis* genetic micro variants with high rifampicin tolerance and increased growth fitness in patients. The precise mechanisms underlying these phenotypes will require further investigation, but it is intriguing to note that different *M. tuberculosis* lineages have varying liabilities for the development of isoniazid resistance (*Carey et al., 2018*), suggesting that clinical isolates may evolve diverse paths towards phenotypic drug resistance that impact fundamental bacterial physiology and tolerance to other antibiotics.

The wide range of observed rifampicin tolerance, spanning many orders of magnitude confirms findings of experimentally evolved drug tolerance to the laboratory isolate *M. tuberculosis*-H37Rv (*Torrey et al., 2016*) and extends our prior findings from a smaller-scale pilot study (*Vijay et al., 2021*). Given that almost all rifampicin resistance is via mutations in *rpoBZaw et al., 2018*, our findings suggest that first-line molecular testing for rifampicin susceptibility, which is replacing phenotypic drug susceptibility (*Macedo et al., 2018*), may not fully capture the response to therapy. It needs to be further validated if these strains that are 'hyper-tolerant' to rifampicin are risk factors for poor clinical outcomes in IR-TB (*Thai et al., 2018*).

Given the association of IR with the emergence of rifampicin resistance (*Srinivasan et al., 2020*), our findings suggest a plausible mechanism by which isoniazid resistance, via rifampicin tolerance, acts as a 'stepping stone' to rifampicin resistance. The association between IR and rifampicin tolerance only held for fast-growing recovered bacteria. Given the observation that 'growing' rifampicin tolerant bacteria are over-represented after initiation of drug therapy in humans due to the specific regulation of *rpoB* in mycobacteria in response to rifampicin exposure (*Zhu et al., 2018*), this may represent a divergence between growing and non-replicating persister forms of antibiotic tolerance. A better understanding of which forms of tolerance contribute to clinically relevant responses to therapy will be critical for tailoring individualized regimens for TB or improving treatment regimens for IR-TB (*WHO, 2018*).

Our study has some limitations. We only assayed rifampicin tolerance under one standard axenic culture condition. It is known that antibiotic tolerance phenotypes vary considerably according to culture conditions (*Hicks et al., 2018*), with some phenotypes only emerging in vitro with specialized media, e.g., containing odd-chained fatty acids (*Hicks et al., 2018*). Second, contributors to antibiotic tolerance can be genetic, epigenetic, or transient (*Su et al., 2016*; *Torrey et al., 2016*; *Hicks et al., 2018*; *Wang et al., 2020*), and there is considerable epistasis between genetic variation and antibiotic susceptibility. Our assay will not be able to capture drivers of tolerance that have been lost in the collection, banking, freezing, and reviving of the *M. tuberculosis* isolates. Finally, the isolates were from a previous study (*Thai et al., 2018*), and during the study period, the old 8 month TB treatment regimen lacked rifampicin in the continuation phase (*Thai et al., 2018*).

This study also reveals interesting aspects like fast and slow-growing sub-populations and possible variation in lag-time distribution among clinical *M. tuberculosis* isolates. There can also be different mechanisms of tolerance and persistence among *M. tuberculosis* isolates, detailed investigations are required to further understand these aspects and its clinical relevance.

In conclusion, our study identifies a significant association between isoniazid resistance and rifampicin tolerance in clinical isolates of *M. tuberculosis.* Our findings have implications for the requirement to consider heterogeneity in bacterial responses to antibiotics and the emergence of antibiotic-tolerant bacterial genetic micro variants in determining optimal tuberculosis treatment regimens.

## Methods
### Ethical approval

*M. tuberculosis* isolates in this study were a part of the collection from a previous study (*Thai et al., 2018*), approved by the Institutional Research Board of Pham Ngoc Thach Hospital as the supervisory institution of the district TB Units (DTUs) in southern Vietnam, Ho Chi Minh City Health Services and the Oxford University Tropical Research Ethics Committee (Oxtrec 030–07).

## Bacterial isolates

242 *M. tuberculosis* isolates, collected for a previous study in Vietnam were used in this study (*Thai et al., 2018*). All the isolates were cultured in the biosafety level-3 laboratory at the Oxford University Clinical Research Unit, Ho Chi Minh City, Vietnam (*Thai et al., 2018*).

## Rifampicin killing assay

Most-probable number-based rifampicin killing assay was done for the clinical *M. tuberculosis* isolates as per the published protocol (*Vijay et al., 2021*). *M. tuberculosis* isolates, after a single sub-culture from the archive, were inoculated in 7H9T medium (Middlebrook 7H9 broth supplemented with 0.2% glycerol, 10% OADC, and 0.05% Tween-80) and incubated at 37 °C until exponential phase with $OD_{600}$ range of 0.4–0.6 is reached. All cultures were homogenized by vortexing for 3 min with sterile glass beads and diluted to the $OD_{600}$ of 0.4. The diluted culture was used for measuring the initial viable bacterial number by the most probable number (MPN) method, using 96 well plates according to the published protocol (*Vijay et al., 2021*). Briefly, the protocol was as follows, a 1 mL aliquot of *M. tuberculosis* culture was harvested, and the cell pellet was washed once. This washed culture was resuspended in 1 mL culture and 100 μL was transferred to 96-well plates as an undiluted culture in duplicate for serial dilution. The undiluted culture was used for 10-fold serial dilution of up to $10^9$ dilutions in microtiter plates (*Figure 1B*). Immediately, after sampling for initial MPN (day 0), the remaining culture in the tube was treated with rifampicin (Merck-Sigma Aldrich, USA) at a final concentration of 2 μg/mL and incubated. On 2 and 5 days post-rifampicin treatment, the viable bacterial number was determined again by the MPN method as previously mentioned (*Vijay et al., 2021*; *Figure 1B*). The growth in 96-well plates was recorded as images by the Vizion image system (Thermo Fisher, Scientific Inc, USA) after 15, 30, and 60 days of incubation, beyond 60 days of drying of plates were observed (*Figure 1B*). The number of wells with visible bacterial growth was determined by two independent readings from two individuals, discrepancies between the two readings were verified and corrected by a third-person reading. MPN value was calculated as mean MPN/mL. The survival fraction at 2 and 5 days post rifampicin treatment was calculated as compared to the initial MPN taken as 100% survival.

## Relative growth difference calculation from MPN number

For calculating the relative growth difference of isolates before rifampicin treatment, the $log_{10}$ MPN ratio between 15 and 60 days of incubation was taken to determine the relative proportion of fast and slow growing sub-populations. A $log_{10}$ MPN ratio close to 0 indicated less growth heterogeneity in the population, whereas a ratio less than 0 indicated the presence of growth heterogeneity due to the presence of fast and slow growth, or heterogeneity in the lag time distribution of sub-populations.

## Drug susceptibility testing

Microtiter drug susceptibility testing was performed using UKMYC6 plates (Thermo Fisher, Scientific Inc·, USA) for determining initial rifampicin and isoniazid phenotypic susceptibility (*Rancoita et al., 2018*). Briefly, three weeks-old *M. tuberculosis* colonies from Lowenstein-Jensen medium were used to make a cellular suspension in 10 mL saline-Tween80 tube with glass beads (Thermo Fisher, Scientific Inc·, USA) and adjusted to 0.5 McFarland units. The suspension is diluted in 7H9 broth (Thermo Fisher, Scientific Inc, USA) and inoculated into a 96-well microtiter plate using a semi-automated Sensititre Autoinoculator (Thermo Fisher, Scientific Inc, USA). Plates were sealed with plastic sheets and incubated at 37 °C for 14–21 days. The minimum inhibitory concentration (MIC) was measured by a Sensititre Vizion Digital MIC Viewing system (Thermo Fisher, Scientific Inc, USA) and considered valid if there was growth in the drug-free control wells. The clinical-resistant cut-off concentrations for isoniazid and rifampicin were 0.1 and 1 μg/mL, respectively.

The IR isolates were also confirmed using the BACTEC MGIT 960 SIRE Kit (Becton Dickinson) according to the manufacturer's instruction in the biosafety level-3 laboratory at the Oxford University Clinical Research Unit (*Thai et al., 2018*). Phenotypic DST was done for streptomycin (1.0 μg/mL), isoniazid (0.1 μg/mL), rifampicin (1.0 μg/mL), and ethambutol (5.0 μg/mL) (*Thai et al., 2018*). Whole genome sequence data was available for the isolates from previously published study (*Srinivasan et al., 2020*) and the Mykrobe predictor TB software platform was used for genotypic antibiotic susceptibility determination of *M. tuberculosis* isolates (*Bradley et al., 2015*).

## Statistical analysis

MDK90 values, and its credible interval was estimated using a linear mixed effect model with a Bayesian approach (brm function, brms package). We used the linear mixed effect model for survival analysis as the data consists of repeated measurements at specific time points. For the linear mixed effect model with the bacterial strains as a random effect, we use the Bayesian approach with non-informative priors, which is equivalent to the frequentist approach. The fixed effect relates to the explanatory variable we are utilizing to predict the outcome. Specifically, our outcome measure is the $\log_{10}$ survival fraction. The explanatory variables encompass isoniazid susceptibility (categorized as isoniazid susceptible or resistant), the day of sample collection (0, 2, and 5 days), and the duration of incubation (15 and 60 days).

Wilcoxon rank-sum test (stat_compare_means function, ggpubr package) was used to test the null hypothesis that the IS and IR groups have the same continuous distribution, as it is a non-parametric test that does not require a strong assumption about the normality of the distribution of the variable. Chi-Square test (odds ratio function, epi tools package) was used to determine if there is a significant relationship between IR and rifampicin tolerance. The Cochran Armitage test (CochranArmitageTest function, DescTools package) was performed to test for trends in IR proportion across the levels of rifampicin tolerance. Linear regression (lm function, stats package) was used to evaluate the correlation between rifampicin survival fraction and relative growth.

Statistical analyses and graphs were plotted using R, version 4·0·1, (*R Development Core Team, 2012*) and p-values of ≤0·05 were considered statistically significant.

## MDK$_{90,}$ $_{99,}$ and $_{99.99}$ calculation

In addition to MDK90 calculated by linear mixed effect model, we also determined the MDK values at 90, 99, and 99.99% reduction in survival fractions for all the *M. tuberculosis* isolates by the following method. The $\log_{10}$ MPN values at Day 0, Day 2, and Day 5 were used to calculate the respective MDK time for 90%, 99%, and 99.99% reduction in fraction of survival. The calculation of MDK time for individual isolates was based on modelling the kill curve as two similar triangles and using the basic proportionality theorem as shown in the flow chart (*Figure 2—figure supplement 3*) to determine the different lengths of the X-axis (Days post rifampicin treatment) corresponding to decline in survival fraction in Y-axis for each MDK time (MDK$_{90,}$ $_{99,}$ and $_{99.99}$).

For example, in case of MDK90, Y0 (MPN number at day 0), Y2 (MPN number at day 2), and Y5 (MPN number at day 5).

First condition tested is, if a 90% reduction in survival fraction happened before or at day 2 by checking if the $\log_{10}$ MPN number on day 2 is less than or equal to a 90% reduction as compared to Y0. If the condition is true then the MDK is calculated as x-axis length DF in the two similar triangles modelled in A (triangles ACB and AFD) and the corresponding formula for X is given below. If the first condition is false then two similar triangles are modelled as in B (triangles ABC and DEC) and X is calculated as 5 – EC. Similarly, for MDK$_{99}$ and MDK$_{99.99}$ time are calculated by applying the condition for 99% and 99.99% reduction in survival fraction.

## Single nucleotide polymorphism difference between longitudinal isoniazid-resistant isolates with differences in rifampicin tolerance

We used whole genome sequence data and genetic variants analysis previously published for identifying non-synonymous single nucleotide polymorphisms (SNPs) emerging in longitudinal isolates from the same patients associated with changes in rifampicin tolerance between the isolates (*Srinivasan et al., 2020*).

## Acknowledgements

We acknowledge funding from the Wellcome Trust Intermediate Fellowship in Public Health and Tropical Medicine to NTTT (206724/Z/17/Z), the Wellcome Trust Investigator Award (207487 /C/17/Z), and NIAID award (R21AI169005) to BJ and Wellcome Trust Major Overseas Program Funding to GT (106680/B/14/Z). We acknowledge Prof. Rosalind Allen (Professor for Theoretical Microbial Ecology at Friedrich Schiller University of Jena), for reading the manuscript and suggestions.

# Additional information

## Funding

| Funder | Grant reference number | Author |
| --- | --- | --- |
| Wellcome Trust | 10.35802/206724 | Nguyen Thuy Thuong |
| Wellcome Trust | 10.35802/207487 | Babak Javid |
| Wellcome Trust | 10.35802/106680 | Guy E Thwaites |
| National Institute of Allergy and Infectious Diseases | R21AI169005 | Babak Javid |

The funders had no role in study design, data collection and interpretation, or the decision to submit the work for publication. For the purpose of Open Access, the authors have applied a CC BY public copyright license to any Author Accepted Manuscript version arising from this submission.

## Author contributions

Srinivasan Vijay, Conceptualization, Data curation, Formal analysis, Supervision, Validation, Investigation, Methodology, Writing – original draft, Writing – review and editing; Nguyen Le Hoai Bao, Conceptualization, Data curation, Formal analysis, Supervision, Investigation, Visualization, Methodology, Writing – original draft, Writing – review and editing; Dao Nguyen Vinh, Formal analysis, Visualization, Writing – review and editing; Le Thanh Hoang Nhat, Formal analysis, Methodology, Writing – review and editing; Do Dang Anh Thu, Nguyen Le Quang, Le Pham Tien Trieu, Hoang Ngoc Nhung, Data curation, Formal analysis, Writing – review and editing; Vu Thi Ngoc Ha, Formal analysis, Writing – review and editing; Phan Vuong Khac Thai, Dang Thi Minh Ha, Resources, Data curation, Writing – review and editing; Nguyen Huu Lan, Resources, Writing – review and editing; Maxine Caws, Guy E Thwaites, Conceptualization, Resources, Supervision, Writing – review and editing; Babak Javid, Conceptualization, Resources, Supervision, Funding acquisition, Visualization; Nguyen Thuy Thuong, Conceptualization, Resources, Data curation, Formal analysis, Supervision, Funding acquisition, Validation, Investigation, Visualization, Project administration, Writing – review and editing

## Author ORCIDs

Srinivasan Vijay ⓘ https://orcid.org/0000-0003-3434-5608
Guy E Thwaites ⓘ https://orcid.org/0000-0002-2858-2087
Nguyen Thuy Thuong ⓘ https://orcid.org/0000-0001-8733-692X

Reviewer #3 (Public review): https://doi.org/10.7554/eLife.93243.3.sa1
Author response https://doi.org/10.7554/eLife.93243.3.sa2

# Additional files

## Supplementary files
- MDAR checklist
- Supplementary file 1. Supplementary material for rifampicin MIC and genetic variant analysis.
- Supplementary file 2. Raw data used for figures.
- Supplementary file 3. Details regarding whole genome sequence data accession number.

## Data availability

All data generated or analysed during this study are included in the supporting files, source data file have been provided for the Figures 1 to 5 as a single file. Isoniazid resistant isolates sequence data has been uploaded ENA under accession number PRJEB78540.

The following dataset was generated:

| Author(s) | Year | Dataset title | Dataset URL | Database and Identifier |
|---|---|---|---|---|
| Vijay S, Bao NLH, Vinh DN, Nhat LTH, Thu DDA, Quang NL, Trieu LPT, Nhung HN, Ha VTN, Thai PVK, Ha DTM, Lan NH, Caws M, Thwaites GE, Javid B, Thuong NT | 2024 | Rifampicin tolerance and growth fitness among isoniazid-resistant clinical Mycobacterium tuberculosis isolates: an in-vitro longitudinal study | https://www.ebi.ac. uk/ena/browser/view/ PRJEB78540 | European Nucleotide Archive, PRJEB78540 |

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
