## [Editor Report · eLife assessment]

This **valuable** study demonstrates that there is significant variation in the susceptibility of isoniazid-resistant Mycobacterium tuberculosis clinical isolates to killing by rifampicin, in some cases at the same tolerance levels as bona fide resistant strains. The evidence provided is **solid**, with no clear genetic marker for increased tolerance, suggesting that there may be multiple routes to achieving this phenotype. The work will be of interest to infectious disease researchers.

---

## [Referee Report · Reviewer #3 (Public review)]

Summary:

The authors have initiated studies to understand the molecular mechanisms underlying the devolvement of multi drug resistance in clinical Mtb strains. They demonstrate the association of isoniazid resistant isolates by rifampicin treatment supporting the idea that selection of MDR is a microenvironment phenomenon and involves a group of isolates.

Strengths:

The methods used in this study are robust and the results support the authors claims to a major extent.

The language has now been corrected and is better to comprehend.

---

## [Author Response]

The following is the authors’ response to the original reviews.

**Reviewer #1 (Public Review):**
Summary:The study entitled "Rifampicin tolerance and growth fitness among isoniazid-resistant clinical Mycobacterium tuberculosis isolates: an in-vitro longitudinal study" by Vijay et al. provides valuable insights into the association of rifampicin tolerance and growth fitness with isoniazid resistance among clinical isolates of M. tuberculosis. Antibiotic tolerance in M. tuberculosis is an important topic since it contributes to the lengthy and complicated treatment required to cure tuberculosis disease and may portend the emergence of antibiotic resistance. The authors found that rifampicin tolerance was correlated with bacterial growth, rifampicin minimum inhibitory concentrations, and isoniazid-resistance mutations.Strengths:The large number of clinical isolates evaluated and their longitudinal nature during treatment for TB (including exposure to rifampin) are strengths of the study.Weaknesses:Some of the methodologies are not well explained or justified and the association of antibiotic tolerance with growth rate is not a novel finding. In addition, the molecular mechanisms underlying rifampicin tolerance only in rapidly growing isoniazid-resistant isolates have not been elucidated and the potential implications of these findings for clinical management are not immediately apparent.

We thank the reviewer for the comments, we have modified the method section and figure 1 to clarify the method as suggested by the reviewer.

Although we agree that previous studies have shown the association of slow growth rate with antibiotic tolerance, ours is the most comprehensive assessment of rifampicin tolerance among clinical isolates, to our knowledge. In particular, we show that the degree of tolerance in clinical isolates can vary over several orders of magnitude: which had not been previously documented or appreciated. Furthermore, the association of high tolerance among IR isolates is a new finding, and given the potential for tolerance to increase risk of de novo drug resistance, our study suggests that IR isolates with high rifampicin tolerance may present a risk for development of MDR-TB.

In addition, we have also analysed the longitudinal isolates and the genetic variants emerging in them associated with increase in rifampicin tolerance. This analysis reveals possible multiple pathways to increase in rifampicin tolerance among clinical M. tuberculosis isolates. Possible clinical implication includes associating high rifampicin tolerance and isoniazid resistance as a risk factor for tuberculosis treatment failure. This study helps to develop further clinical studies to evaluate the role of rifampicin tolerance in IR isolates and treatment outcome. We have focused on these aspects in the discussion of the revised manuscript.

**Reviewer #2 (Public Review):**
Summary:This study by Vijay and colleagues addresses a clinically important, and often overlooked aspect of Tb treatment. Detecting for variations in the level of antibiotic tolerance amongst otherwise antibiotic-susceptible isolates is difficult to routinely screen for, and consequently not performed. The authors, present a convincing argument that indeed, there is significant variation in the susceptibility of isoniazid-resistant strains to killing by rifampicin, in some cases at the same tolerance levels as bona fide resistant strains. On the whole, the study is easy to follow and the results are justified. This work should be of interest to the wider TB community at both a clinical and basic level.Weaknesses:The manuscript is long, repetitive in places, and the figures could use some amending to improve clarity (this could be a me-specific issue as they look ok on my screen, yet the colour is poor when printed).

We thank the reviewer for the comments, we have modified the revised manuscript as per the reviewer suggestions.

It would have been great to have seen some correlation between increased rifampicin tolerance and treatment outcome, although I'm not sure if this data is available to the researchers. I agree with the researchers the use of a single media condition is a limitation. However, this is true of a lot of studies. Rifampicin tolerance and treatment outcome analysis.

We agree with the reviewer that correlation between rifampicin tolerance and treatment outcome is important. This needs to be performed in future studies with better design to correlate rifampicin tolerance with treatment progression or outcome data.

**Reviewer #3 (Public Review):**
Summary:The authors have initiated studies to understand the molecular mechanisms underlying the devolvement of multi-drug resistance in clinical Mtb strains. They demonstrate the association of isoniazid-resistant isolates by rifampicin treatment supporting the idea that selection of MDR is a microenvironment phenomenon and involves a group of isolates.Strengths:The methods used in this study are robust and the results support the authors' claims to a major extent.Weaknesses:The manuscript needs a thorough vetting of the language. At present, the language makes it very difficult to comprehend the methodology and results.

We thank the reviewer for the comments, we have revised the manuscript as per the reviewer’s suggestions.

**Reviewer #1 (Recommendations For The Authors):**
Major comments:(1) Methods: The authors attempt to differentiate between "fast"- and "slow"-growing bacteria in order to determine if the growth rate is associated with rifampicin tolerance. This is accomplished by assessing growth on solid agar at 15 and 60 days post-incubation, respectively. However, mycobacterial growth rate is not a binary phenomenon but rather a continuous variable. Moreover, it is not clear why 15 and 60 days were selected. Also, instead of a "slow growth" phenotype, the 60-day time point might simply reflect a longer lag phase. Were the plates examined at any interval time points? It would be interesting to know whether colony growth was delayed overall in the populations observed only at 60 days, or simply if the appearance of microcolonies visible to the naked eye was delayed (with normal growth afterwards).

We thank the reviewer for the comments, we want to clarify that we have not used agar plates but most-probable number method to determine the survival fraction post antibiotic treatment. We have clarified this in the revised manuscript and revised figure 1. The MPN method is a binary measure (growth/ no growth) and therefore cannot differentiate between long lag time and other mechanisms. In our original analysis, we included an intermediate time point of 30 days, but these data (included as supp fig. 1) cannot address the issue of lag phase directly. Since the 30-day time point did not add to the overall analysis and interpretation, we had not included them in the original submission.

(2) Methods/Results/Discussion: Some important clinical information is missing-how were the patients treated who had IR isolates? Did they receive the standard regimen for DS TB or was another drug substituted for isoniazid? Exposure to different drugs could affect the rifampicin-tolerant populations during the intensive phase (Figure 5).

Thank you for this comment, we have included the information regarding the treatment regimen in the revised manuscript.

Were there differences in microbiological (sputum culture conversion rate at 8 weeks or time to culture negativity) or clinical outcomes based on isoniazid susceptibility? Perhaps more importantly, were there differences in microbiological/clinical outcomes based on the proportion of bacterial subpopulations with rifampicin tolerance for a particular isolate? There should be more discussion on the potential clinical implications of the study's findings.

We agree with the reviewer that correlation between rifampicin tolerance and treatment progression or outcome is important. This needs to be performed in future studies with better design to correlate rifampicin tolerance with treatment progression or outcome data.

(3) Results (Figure 3A): Although an interesting finding, the increased rifampicin tolerance observed only in the "rapidly" growing populations of isoniazid-resistant isolates (IR) vs. isoniazid-susceptible (IS) isolates is not explained. In contrast, equally, increased rifampicin tolerance is seen in the "slowly" growing populations of both IR and IS isolates. It would be interesting to know if these slowly growing populations show specific tolerance to rifampicin or if, as expected, slow growth confers tolerance to a range of different bactericidal antibiotics.

We thank the reviewer for the suggestions. we agree these will be interesting to investigate in a future study but are outside the scope of the current study.

(4) Results (Figure 3B): The basis for the classification into tertiles is not clear and appears somewhat arbitrary-does this represent the survival of a particular isolate following rifampicin exposure relative to the other isolates based on isoniazid susceptibility (IS or IR) or the % growth relative to other populations for the same isolate? Figure 3B is missing a y-axis label. Is it a log10 MPN ratio?

We thank the reviewer for pointing this, we want to clarify that for the classification into tertiles, first we pooled both group of isolates isoniazid susceptible (IS) and isoniazid resistant (IR) into a single population. Subsequently, we categorized this unified population into three distinct groups: low, medium, and high, based on their survival fraction following rifampicin treatment. Consequently, the 'low,' 'medium,' and 'high' tertiles represent the survival of each isolate following rifampicin exposure relative to the total number of isolates combing both IS and IR isolates.

For clarity, we provide a breakdown of the criteria for each tertile:

+Low tertile: Consists of isolates with the lowest survival fraction (bottom 25%).

+Medium tertile: Encompasses isolates with survival fractions that fall between the bottom 25% and the top 25%.

+High tertile: Comprises isolates with the highest survival fractions (top 25%). This we have modified in the revised manuscript to clarify.

We have also modified the Figure 3B to correct the y-axis label.

(5) Results (lines 185-186): For correlating relative growth in the absence of antibiotics, 19 clinical isolates "outliers" were removed without explanation.

We have added explanation for the “outliers” which were removed earlier due to deviation from normal distribution, we have also provided the supplementary figure 3 which includes these outliers.

(6) Results (lines 203-211): The authors attempted to investigate a potential association between the mechanism of M. tuberculosis isoniazid resistance and the degree of rifampicin tolerance. However, the vast majority of IR clinical isolates (n=71) had a katG_S315X mutation and only 8 isolates had alternative mutations (inhA_I21T and fabG1_C-15X). Given the wide range of rifampicin tolerance observed within these isoniazid-resistant isolates, they concluded that other genetic or epigenetic determinants must be playing a role. WGS of longitudinally collected isolates from the same patients during TB treatment yielded non-synonymous SNPs in a list of genes previously reported to be associated with persistence, tolerance, and mycobacterial survival. However, precise mechanisms (including, e.g., expression of efflux pumps) are not investigated.

We thank the reviewer for summarising the findings. Yes, we agree that investigating the precise mechanism of rifampicin tolerance is beyond the scope of the current work.

Minor comments:(1) Abstract (line 41): The nonstandard abbreviations "IR" and "IS" have not been introduced prior to this usage.

We have modified this in the abstract.

(2) Introduction (line 60): Insert "phenomena" or "mechanisms" after "two".

We have modified this in the introduction.

(3) Introduction (lines 66-69): This sentence is confusing, especially the second part ("supporting this studies...").

We have modified the lines to clarify.

(4) Introduction (line 84): In the current text, it appears as if "IR" is the abbreviation for "isoniazid". Therefore, I recommend changing "resistance to isoniazid" to "isoniazid resistance".

We have modified this in the revised manuscript.

(5) Results (line 141): Insert "the" before "rest".

We have modified this in the revised manuscript.

(6) Results (line 187): Replace "did not had" with "did not have".

We have modified this in the revised manuscript.

**Reviewer #2 (Recommendations For The Authors):**
Abstract:The abstract is long and repetitive. It needs reworking and shortening to improve clarity and highlight the main takeaway message.

We thanks the reviewer for the suggestions and have modified this in the revised manuscript.

The introduction is interesting and contains relevant information. However, it is long and takes a while to get to the point of the study. It needs re-writing to emphasise key prior results and the purpose of this study.

We thanks the reviewer for the suggestions and we have modified this in the revised manuscript.

Results:As the study relies predominately on the use of MPN, I think a simple schematic of how the experiment is performed would be informative. Could this be added to Figure 1?

We have revised the figure 1 in the manuscript to include the schematic representation.

Some of the differences in MKD90, whilst they may be significant, are small so it would at least provide context as to the relevance of these differences. This may also alleviate my confusion as to how the authors can measure the time required to achieve MDK90 as 1.23-1.31 days when the first time point that is taken is day 2 (the data in Figure 2). They have FigS6 but this is small and hard to follow.

We thank the reviewer for this suggestion, we have modified this in the revised manuscript and figureS6.

Figure 2:Would be helpful to have -1 on the Y axis.The grey dots don't print very well (Might be my printer)

We have modified this in the revised manuscript, figure 2.

Line 142: The authors note a difference in RIF tolerance at day 15 that disappeared by day 60. I assume they are referring to the day 5 timepoint although this isn't clear as written.

Yes, it is referring to the day 5 time point and we have clarified this in the revised manuscript.

The section starting at line 148 (fig 3) is interesting, but it is difficult to read and follow what the difference is between this data and the prior data in Figure 2. It also wasn't until about line 165 that the purpose became clear. Overall the conclusions are sound and interesting.

We have modified this in the revised manuscript.

Line 154: What are the early and late time recovery time points?Is Figure 3A the same data as Figure 2?

We have clarified this in the revised manuscript, the figure 3A is the same data as Figure 2.

I found Figure 6 hard to follow. I'm not sure how better to present this data, but it should be improved. Some further clarification in the text would be helpful.

We thank the reviewer for the suggestions. We have added more explanation in the text to clarify figure 6.

Conclusions:The conclusions are sound, based on the data presented. The clinical relevance is highlighted, yet appropriately phrased to not be too far-reaching.Again, I think the conclusions could be condensed considerably. It is repetitive in places, which distills the main outcomes of this otherwise interesting and important study. The authors appropriately highlight some of the limitations of their study.

We thank the reviewer for these comments and have modified this in the revised manuscript.

**Reviewer #3 (Recommendations For The Authors):**
The manuscript "Rifampicin tolerance and growth fitness among isoniazid-resistant clinical Mycobacterium tuberculosis isolates: an in-vitro longitudinal study" by Srinivasan et.al., details the identification/ development of isoniazid-resistant strains in clinical isolates following testament with rifampicin. This is an important aspect of understanding MDR development in TB strains. the results are promising and gel well with the hypothesis. However, the manuscript requires a thorough language modification. While the overall idea is clear the methodology does not come out clearly.Specific comments:(1) It is not clear whether rifampicin treatments were given for 2 and 5 days before kill curves or for 15 and 60 days? The methodology needs to be phased clearly. Why was this time interval of 15 days and 60 days taken? is there a rationale for this?

We thank the reviewer for the suggestions, we have modified the method and figure 1 to clarify this in the revised manuscript.

(2) A concentration of 2ug/ml was used for in vitro culture in this study. While the authors themselves indicate that this is well above the MIC, this might represent a non- natural dose and hence may force the evolution of strains. What will be the scenario in the natural course of antibiotic treatment (dose at MIC or less than MIC)?

We have observed that till 5 days there is no significant resistant emergence but after 5 days only resistance emerges, therefore we avoided determining the survival fraction after resistance emergence, the kill curve represents mostly tolerant sub population. ADD: Pharmacokinetic studies of rifampicin dosing suggest that peak concentrations of >2-32 µg/mL are typical for standard doses of the drug, therefore we believe the chosen concentration of 2 µg/mL to be physiologically relevant.

(3) As described in line 155, the survival spanned a broad distribution, across a million times in difference. This is rather surprising that 5 days of rifampicin treatment would lead to such a spread in resistance patterns. Did the authors study the different populations to understand this phenomenon? This is important given the scale of resistance developed in this short time.

We want to clarify that the broad range of survival fraction reflect the difference in tolerant sub-populations but not resistant sub-population to rifampicin as they are determined post rifampicin treatment in rifampicin free media, this has been clarified in the revised figure 1.

Overall, the manuscript is a detailed study with new insights into the development of multi-drug resistance by Mtb. A thorough vetting for language is essential for a greater impact of the study.

We thank the reviewer and have attempted to improve the clarity of the language to increase the potential impact of our findings.